Bioinformatics analysis of laryngeal squamous cell carcinoma: seeking key candidate genes and pathways

Ma Jinhua
Hu Xiaodong
Dai Baoqiang
Wang Qiang
Wang Hongqin hongqinwang0317@126.com
Department of Otolaryngology, Cangzhou Central Hospital , Cangzhou, Hebei , China
Choudhary Kumari Sonal
Electronic publication date: 2021 Apr 14
Publication date: 2021
Volume: 9
Electronic Location ID: e11259
Received 2020 Nov 26; Accepted 2021 Mar 22
Copyright: © 2021 Ma et al.
Copyright year: 2021
Copyright holder: Ma et al.
License: This is an open access article distributed under the terms of the Creative Commons Attribution License, which permits unrestricted use, distribution, reproduction and adaptation in any medium and for any purpose provided that it is properly attributed. For attribution, the original author(s), title, publication source (PeerJ) and either DOI or URL of the article must be cited.
License URL: https://creativecommons.org/licenses/by/4.0/

Keywords: Laryngeal squamous cell carcinoma, Bioinformatics analysis, GO, KEGG

Funding: Hebei Medical Research Youth Program 20200175 This work was supported by the Hebei Medical Research Youth Program (20200175). The funders had no role in study design, data collection and analysis, decision to publish, or preparation of the manuscript.

==============================
Background

Laryngeal squamous cell carcinoma (LSCC) is the second most aggressive head and neck squamous cell carcinoma. Although much work has been done to optimize its treatment, patients with LSCC still have poor prognosis. Therefore, figuring out differentially expressed genes (DEGs) contained in the progression of LSCC and employing them as potential therapeutic targets or biomarkers for LSCC is extremely meaningful.

Methods

Overlapping DEGs were screened from two standalone Gene Expression Omnibus datasets, and Gene Ontology and Kyoto Encyclopedia of Genes and Genomes pathway enrichment analyses were performed. By applying STRING and Cytoscape, a protein–protein network was built, and module analysis was carried out. The hub genes were selected by maximal clique centrality with the CytoHubba plugin of Cytoscape. UALCAN and GEPIA data were examined to validate the gene expression findings. Moreover, the connection of the hub genes with LSCC patient overall survival was studied employing The Cancer Genome Atlas. Then, western blot, qRT-PCR, CCK-8, wound healing and transwell assays were bring to use for further verify the key genes.

Results

A total of 235 DEGs were recorded, including 83 upregulated and 152 downregulated genes. A total of nine hub genes that displayed a high degree of connectivity were selected. UALCAN and GEPIA databases verified that these genes were highly expressed in LSCC tissues. High expression of the SPP1, SERPINE1 and Matrix metalloproteinases 1 (MMP1) genes was connected to worse prognosis in patients with LSCC, according to the GEPIA online tool. Western blot and qRT-PCR testify SPP1, SERPINE1 and MMP1 were upregulated in LSCC cells. Inhibition of SPP1, SERPINE1 and MMP1 suppressed cell proliferation, invasion and migration.

Conclusion

The work here identified effective and reliable diagnostic and prognostic molecular biomarkers by unified bioinformatics analysis and experimental verification, indicating novel and necessary therapeutic targets for LSCC.

Introduction

As the second most aggressive head and neck squamous cell carcinoma, the incidence of laryngeal squamous cell carcinoma (LSCC) is rapidly increasing, and in 2018, approximately 13,150 new cases were expected to be found in the United States, of which approximately 3,710 would die due to the disease (Jemal et al., 2007; Patel et al., 2019). Although good outcomes can be achieved for those with LSCC through the use of accurate surgery and chemoradiotherapy, 30% of patients experience disease recurrence or distant metastasis that can lead to death (Johnson et al., 2019). Moreover, the mortality of patients with more advanced LSCC tumours (stage III and IV) is higher (Hermida-Prado et al., 2019). So it is significant and urgent to identify key biomarkers and novel therapeutic targets in LSCC.

Cancer is a heterogeneous disease that is characterized by many kinds of gene aberrations, and this is also the case for LSCC (Shen et al., 2019). However, the mechanisms of LSCC development are not completely understood. With the continuous advances in microarray technology and bioinformatics analysis, gene chip technology plays a significant role in exploring tumour gene expression profiles and identifying the differentially expressed genes (DEGs) and functional pathways associated to tumorigenesis and prognosis (Tinker, Boussioutas & Bowtell, 2006). In this work, the microarray data of two gene expression profiles were obtained, and DEGs were recognized between cancerous and noncancerous tissues, succeeded by deeper evaluation with Gene Ontology (GO) and Kyoto Encyclopedia of Genes and Genomes (KEGG) pathway analyses. Moreover, a protein–protein interaction (PPI) network of DEGs was constructed, and the CytoHubba plugin of Cytoscape was used to identify the hub genes. Additionally, the overlapping gene expression between healthy and tumour tissues was confirmed by the UALCAN and GEPIA online databases. The following nine DEGs were selected for further analysis: Matrix metalloproteinases 9 (MMP9), SPP1, SERPINE1, MMP1, MMP13, MMP3, CXCL8, OSM and COL1A1. GEPIA was played to evaluate the prognosis of the mentioned hub genes with disease prognosis and showed that SPP1, SERPINE1 and MMP1 were correlated with worse survival. Then, we demonstrated by western blot and qRT-PCR that SPP1, SERPINE1 and MMP1 were upregulated in LSCC cells. Inhibition of SPP1, SERPINE1 and MMP1 could suppress cell proliferation, invasion and migration. In summary, the bioinformatic study presented here offers some promising biomarkers connected to the development and prognosis of patients with LSCC.

Materials and Methods

DEGs identification in LSCC

We obtained the gene expression profiles of LSCC from the Gene Expression Omnibus (GEO) database (https://www.ncbi.nlm.nih.gov/geo/) (Barrett et al., 2013). The raw gene expression profile datasets GSE59102 and GSE107591 were obtained from that database. The platform for GSE59102 is GPL6480, Agilent-014850 Human Genome Microarray 4 × 44 K G4112F (Probe Name version), including 13 normal tissues and 29 tumour tissues. The platform for GSE107591 is GPL6244, HuGene-1_0-st Affymetrix Human Gene 1.0 ST Array, containing 23 normal and 24 tumour tissues. The DEGs between cancerous and noncancerous tissues were detected by GEO2R (http://www.ncbi.nlm.nih.gov/geo/geo2r) with the criteria of logFC (fold change) > 1 and adjusted p-value < 0.01, which were thought to indicate clearly differential expression (Davis & Meltzer, 2007). The next step was to use the online Venn software to identify the intersection of DEGs between the two datasets.

KEGG and GO enrichment analyses of DEGs

We used the Database for Annotation, Visualization and Integrated Discovery (DAVID, http://david.abcc.ncifcrf.gov/) online tool to study the roles of the identified DEGs and put into effect functional and pathway enrichment analysis (Huang et al., 2007). KEGG (http://www.kegg.jp), an integrated database, was applied to determine the high-level functions and the biological interpretation of genome sequences and other high-throughput data (Kanehisa et al., 2016). The GO (http://www.geneontology.org) project is a primary bioinformatics tool for annotating genes and analysing the biological processes (BPs) of the genes and includes the BP, cellular component (CC) and molecular function (MF) (Ashburner et al., 2000). p-value < 0.05 was thought to refer a important difference in statistic.

PPI network construction and module analysis

A PPI network of DEGs was foreseen by the Search Tool for the Retrieval of Interacting Genes/Proteins (STRING) online database (version 11.0; http://string-db.org) (Szklarczyk et al., 2015). Based on the STRING online tool, PPIs of the DEGs were constructed with a confidence score > 0.4, we exported the results as a simple tabular text. Cytoscape software version 3.7.1 (www.cytoscape.org) is an open source bioinformatics software platform for visualizing molecular interaction networks. Subsequently, the PPI network was visualized by means of Cytoscape software (Holmås et al., 2019). The most significant module in that network was detected by the Molecular Complex Detection (MCODE) (version 1.4.2) plug-in of Cytoscape (Bandettini et al., 2012). The selection standards: MCODE score > 5; degree cut-off = 2; node score cut-off = 0.2; max depth = 100; and k-score = 2. Afterward, KEGG and GO analyses of the genes in the PPI were performed by DAVID.

Hub gene selection and analysis

The CytoHubba plug-in of Cytoscape is an application that contains several topological algorithms for ranking nodes in a PPI network by the network characteristics. In this work, maximal clique centrality (MCC) was selected to explore the top nine hub genes among the 12 available computing methods (Chin et al., 2014). Moreover, the UALCAN (http://ualcan.path.uab.edu/) and GEPIA (http://gepia.cancer-pku.cn/) online databases were resorted to validate gene expression. UALCAN and GEPIA are internet tools that provide a straightforward way to investigate publicly available cancer transcriptome data, including The Cancer Genome Atlas (Chandrashekar et al., 2017; Tang et al., 2017). Moreover, the GEPIA database was applied to survival analysis of the hub genes. p-Value < 0.05 was considered as inportant in statistic.

Cell culture and cell transfection

The human LSCC cell lines Hep2 and LSC-1 (Bluefbio, China) and human laryngeal epithelial cells (HLECs; Lifeline, Framingham, MA, USA) were cultured in Dulbecco’s modified Eagle’s medium (DMEM) containing 10% foetal bovine serum, 100 μ/mL streptomycin and 100 μ/mL penicillin, the culture conditions were 37 °C and 5% CO2. All cells were passaged 3–4 times before use (Wang et al., 2018).

The siRNA-SPP1, siRNA-SERPINE1 and siRNA-MMP1 that were transfected into Hep2 and LSC-1 cell lines by Lipofectamine 2000 were obtained from GeneCreate (China) in accordance with the instuctions of manufacturer. Hep2 and LSC-1 cells were cultured in DMEM at least 24 h and before transfection, they also need to washed with phosphate-buffered saline (PBS, pH 7.4) before transfection (Liu, Ren & Song, 2019).

Western blot

Total protein isolated from Hep2 and LSC-1 cells and HLECs were loaded into sodium dodecyl sulfate polyacrylamide gel electrophoresis (SDS-PAGE) gels and transferred to polyvinylidene fluoride (PVDF) membranes. The PVDF was hindered with 5% skim milk at 37 °C for 90 min. Next, the PVDF membrane was incubated with SPP1, SERPINE1 or MMP1 rabbit polyclonal antibodies (Proteintech, Rosemont, IL, USA) and β-actin rabbit polyclonal antibodies (ABclonal, Woburn, MA, USA) at 4 °C overnight. And the PVDF was rinsed with PBST for 10 min and repeated three times. Then, the PVDF was incubated with horseradish peroxidase-labelled goat anti-rabbit IgG secondary antibody (Jackson, MS, USA) at 37 °C for 60 min. The PVDF was rinsed with PBST for 10 min and repeated three times. Then, the immunoactivity was detected by optical luminometry (Mishra, Tiwari & Gomes, 2017).

qRT-PCR

To isolate total RNA from Hep2 cells, LSC-1 cells and HLECs, we used TRIzol reagent (TAKARA, Japan) according to the instructions. The conditions of reverse transcription reaction can be noticed: 25 °C for 10 min, 50 °C for 30 min and 85 °C for 5 min. Detection using the fluorescence quantitative PCR kit. The conditions of fluorescence quantitative PCR were as follows: 95 °C for 5 min, 95 °C for 10 s, 60 °C for 30 s, a total of 40 cycles. The solubility curve temperature range was set at 60–95 °C, and three replicate wells were prepared for each specimen. SPP1, SERPINE1 and MMP1 used β-actin as internal factors (Wilhelm & Pingoud, 2003). The qRT-PCR results were determined by the 2−ΔΔCt approach, and the primer sequences are listed in Table 1.

Table 1 Primer sequences.

Target gene	Primer (5′–3′)	
Hu-Actin-F	CATGTACGTTGCTATCCAGGC	
Hu-Actin-R	CTCCTTAATGTCACGCACGAT	
Hu-SPP1-F	TGTGTTGGTGGAGGATGTC	
Hu-SPP1-R	GCGTTTGGCTGAGAAGG	
Hu-SERPINE1-F	GTTCATTGCTGCCCCTT	
Hu-SERPINE1-R	CCTGGTCATGTTGCCTTT	
Hu-MPP1-F	GAG TAT ATC TGC CAC TCC TTG AC	
Hu-MPP1-R	CTT GGA TTG ATT TGA GAT AAG TCA TAG C	

CCK-8 assay

To determine the proliferation of Hep2 and LSC-1 cells, we used a CCK-8 assay. We incubated the cells in 96-well culture plates and the inoculation density was 3 × 105. For cell transfection, the cells were cultured overnight. After transfection for 48 h, 10 mL of CCK-8 solution was added to each well, and the cells were incubated at 37 °C for another 60 min. A total of 490 nm was used as the absorbance of the solutions, and the absorbance of the solutions was detected by a Smart Microplate Reader 16.1 (Ma et al., 2017).

Transwell migration assay

For cell migration tests of Hep2 and LSC-1 cells, Transwell chambers with a polycarbonate membrane were used. In the upper chambers, Hep2 and LSC-1 cells were incubated in serum-free DMEM, and 10% FBS was out to the lower chambers. After 10 h, the Hep2 and LSC-1 cells in the upper chambers were removed, and the cells in the lower chambers were dyed with crystal violet at 25 °C for 1 min. A light microscope was used to observe and count the cells in five randomly selected fields.

Wound healing assay

Hep2 and LSC-1 cells were inoculated into the 6-well plates. When cells were at 90–100% confluence, wounds of uniform width were created by slowly pulling a 10 μL pipette perpendicular to the bottom of the 6-well plate (three wounds/well). Then, the cells were rinsed with PBS three times to remove floating cells, and incubated in serum-free DMEM to inhibit cell proliferation and division. At 0 and 24 h after wounding and culture, the migration distance of cells in the wound area was observed under a microscope, and several different fields of view were randomly selected for photographing (He et al., 2020). The experiment was repeated three times.

Statistical analysis

Statistical Product and Service Solutions 25.0 software was ran for analysis in statistic. The t-test was applied to numerical data, and p < 0.05 indicated the significance level.

Results

DEGs identification in LSCC

Before identifying DEGs in LSCC, the microarray was standardized. We identified 3,359 DEGs in GSE59102 and 444 DEGs in GSE107591. The overlap between the two datasets included 341 genes (Fig. 1), including 205 downregulated genes and 136 upregulated ones between LSCC tissues and control tissues.

Figure 1 Identification of DEGs in gene expression profile datasets (GSE59102 and GSE107591).

KEGG and GO enrichment analyses of DEGs

Gene Ontology and KEGG pathway enrichment analyses were performed in DAVID (Table 2) to study the biological classification of the DEGs. The BP category outcomes of the GO analysis outcomes expressed that the DEGs were markedly associated with extracellular matrix (ECM) organization, cell adhesion, the collagen catabolism pathway and positive regulation of cell-substrate adhesion. Within the CC categories, the DEGs were markedly linked with the extracellular region, ECM and proteinaceous ECM. Among the MF categories, the DEGs were significantly enriched in heparin binding and ECM structural constituents. In addition, KEGG signalling pathway analysis illustrated that the DEGs were pivotal in the following pathways: ‘ECM-receptor interaction,’ ‘drug metabolism-cytochrome P450,’ ‘focal adhesion,’ ‘PI3K-Akt signaling pathway’ and ‘complement and coagulation cascades.’

Table 2 GO and KEGG pathway enrichment analysis of DEGs in LSCC samples.

Pathway ID	Pathway description	Count	p-Value	
GO:0030198	Extracellular matrix organization	24	2.02E−12	
GO:0007155	Cell adhesion	32	5.83E−10	
GO:0030574	Collagen catabolic process	12	2.37E−08	
GO:0010811	Positive regulation of cell-substrate adhesion	8	5.50E−06	
GO:0030199	Collagen fibril organization	8	6.60E−06	
GO:0005576	Extracellular region	75	2.87E−14	
GO:0031012	Extracellular matrix	29	8.02E−13	
GO:0005578	Proteinaceous extracellular matrix	26	1.92E−11	
GO:0070062	Extracellular exosome	98	4.67E−11	
GO:0016324	Apical plasma membrane	18	2.26E−05	
GO:0008201	Heparin binding	15	1.17E−06	
GO:0005201	Extracellular matrix structural constituent	8	1.94E−04	
hsa04512	ECM-receptor interaction	13	7.22E−08	
hsa00982	Drug metabolism-cytochrome P450	10	4.77E−06	
hsa04510	Focal adhesion	14	1.38E−04	
hsa04151	PI3K-Akt signaling pathway	17	8.00E−04	
hsa04610	Complement and coagulation cascades	6	0.009851474	

PPI network construction and remarkable module identification

The STRING database was used to predict the interactions among the DEGs with a combined score >0.4 at the protein level. The PPI network was built with Cytoscape software and included 176 nodes and 371 edges (Fig. 2A). Additionally, the hubs of the PPI network module were obtained by MCODE and consisted of 9 nodes and 34 edges (Fig. 2B). The functional analyses of genes contained with the modules were identified by running DAVID. The outcomes indicated that the genes of this module were highly abundant in condensed nuclear chromosome outer kinetochore, ATP binding and ATP-dependent microtubule motor activity, plus-end-directed (Table 3). To more deeply investigate the top nine hub genes, we used the cytoHubba plug-in of Cytoscape in the above PPI using the MCC method. The hub genes MMP9, SPP1, SERPINE1, MMP1, MMP13, MMP3, CXCL8, OSM and COL1A1 were used for further analysis. The names, abbreviations and synonyms for these hub genes can be seen in Table 4.

Figure 2 Common DEG PPI network construction and module analysis.

(A) A total of 235 DEGs were visualized in the DEG PPI network complex: the nodes represent proteins, and the edges represent the interactions of the proteins. (B) Module analysis using MCODE: MCODE score > 5; degree cut-off = 2; node score cut-off = 0.2; max depth = 100; and k-score = 2.

Table 3 Functional enrichment analysis of DEGs in the most significant module.

Pathway ID	Pathway description	Count	p-Value	
GO:0000942	Condensed nuclear chromosome outer kinetochore	2	0.001292732	
GO:0005524	ATP binding	4	0.003547288	
GO:0008574	ATP-dependent microtubule motor activity, plus-end-directed	2	0.004812021	
cfa04114	Oocyte meiosis	2	0.016	

Table 4 Abbreviations, official full names and synonyms for the nine hub genes.

Abbreviations	Official full names	Synonyms	
MMP9	Matrix metallopeptidase 9	CLG4B, GELB	
SPP1	Secreted phosphoprotein 1	BNSP, OPN	
SERPINE1	Serpin family E member 1	PLANH1, PAI-1	
MMP1	Matrix metallopeptidase 1	CLG, CLGN	
MMP13	Matrix metallopeptidase 13	Collagenase 3	
MMP3	Matrix metallopeptidase 3	Stromelysin-1, STMY1	
CXCL8	C-X-C motif chemokine ligand 8	IL-8	
OSM	Oncostatin M	Oncostatin-M	
COL1A1	Collagen type I alpha 1 chain	Type I proalpha 1, EDSC	

High expression of SPP1, SERPINE1 and MMP1 in LSCC and their association with poor prognosis

Tools in the UALCAN and GEPIA databases were used to evaluate the expression levels of these nine genes in LSCC. Both databases confirmed that the expression of MMP9, SPP1, SERPINE1, MMP1, MMP13, MMP3, CXCL8, OSM and COL1A1 showed obvious differences in LSCC samples and healthy samples, which was in accordance with the results discussed above (Fig. 3). Using the GEPIA database to study the relationship between gene expression and patient survival, we found that only SPP1, SERPINE1 and MMP1 were obviously connected with the overall survival of patients with LSCC (Fig. 4). Therefore, we verified the expression of SPP1, SERPINE1 and MMP1 in LSCC cells by western blot (Figs. 5A–5D) and qRT-PCR (Fig. 5E). The outcomes indicated that SPP1, SERPINE1 and MMP1 were highly expressed in HEP2 and LSC-1 cells. These results were consistent with the bioinformatics analysis.

Figure 3 The expression level of hub genes between cancerous and noncancerous tissues according to the UALCAN and GEPIA databases.

(A–I) UALCAN database. (J–R) GEPIA database. (*p < 0.05).

Figure 4 The prognostic information of the nine hub genes.

(A–I) The GEPIA online tool was used to identify the prognostic value of the hub genes, and three of nine were correlated with worse survival (p < 0.05).

Figure 5 Expressions of SPP1, SERPINE1 and MMP1 in LSCC cells and HLEC.

(A–D) The proteins levels of SPP1, SERPINE1 and MMP1 were determined by western blotting in HLEC, LSC-1 and Hep2 cells. (E) Relative expression of SPP1, SERPINE1 and MMP1 in HLEC, Hep2 and LSC-1 cells was examined by qPCR and normalized to β-actin expression. (***p < 0.001).

Downregulation of SPP1, SERPINE1 and MMP1 expression inhibited LSCC cell proliferation in vitro

To further verify the function of SPP1, SERPINE1 and MMP1 in HEp-2 and LSC-1 cells, SPP1, SERPINE1 and MMP1 were knocked down in vitro. CCK-8 was performed to verify the effects of SPP1, SERPINE1 and MMP1 on HEp-2 and LSC-1 cell proliferation in vitro. HEp2 and LSC-1 cells treated with siRNA-SSP1, siRNA-SERPINE1, siRNA-MMP1 and siRNA-NC were cultured and then subjected to CCK-8 assays. The results showed that the growth rate of HEp-2 and LSC-1 cells was significantly inhibited after knockdown of SPP1, SERPINE1 and MMP1 (Fig. 6).

Figure 6 Effects of SPP1, SERPINE1 and MMP1 on LSCC cells proliferation in vitro.

(A) Hep2 cells (B) LSC-1 cells. (*p < 0.05, **p < 0.01).

Downregulation of SPP1, SERPINE1 and MMP1 expression inhibited LSCC cell migration and invasion in vitro

Invasion and metastasis are important features of tumours. Therefore, the effects of SPP1, SERPINE1 and MMP1 on migration and invasion of HEp-2 and LSC-1 cells were analysed in depth. First, transwell migration assay was performed, and the results were consistent with those of the wound healing assay. After knockdown of any of these genes, the amount of cells passing through the filter membrane to the lower chamber was decreased significantly (p < 0.05, Figs. 7A–7J). Next, a wound healing assay was used to detect the cell migration capacities in the control group and the SPP1, SERPINE1 and MMP1 knockdown groups. The results showed that the migration distance of cells in the SPP1, SERPINE1 and MMP1 knockdown groups was shorter when comparing with that in control set after 24 h (p < 0.05, Figs. 7K and 7L).

Figure 7 Effects of MMP1, SERPINE1 and SPP1 on LSCC cells migration and invasion in vitro.

(A–E) Hep2 cells were treated with SiRNA-NC, SiRNA-MMP1, SiRNA- SERPINE1 and SiRNA-SPP1, and the effects on cell migration and invasion were determined with cell transwell test. (F–J) LSC-1 cells were treated with SiRNA-NC, SiRNA-MMP1, SiRNA-SERPINE1 and SiRNA-SPP1, and the effects on cell migration and invasion were determined with cell transwell test. (K) The effects on Hep2 cell migration were determined with wound healing assay. (L) The effects on LSC-1 cell migration were determined with wound healing assay (*p < 0.05, **p < 0.01, ***p < 0.001).

Discussion

Tobacco use, alcohol consumption, and rising incidences of viral infections, such as human papillomavirus infection, have been considered the principal aetiological factors of the pathogenesis of LSCC (D’Souza et al., 2017; Gheit et al., 2014). However, the development of LSCC is a sophisticated biological programme. For the past few years, many biomarkers have been applied for the diagnosis and treatment of LSCC (Cossu et al., 2019). Various anti-LSCC mechanisms have also been identified (Chrysovergis et al., 2019). At the same time, few studies have been performed at the multigene level. Studies at the multigene level can contribute to exploring cancer pathogenesis.

In this work, the microarray datasets GSE59102 and GSE107591 were selected to confirm DEGs between cancerous and noncancerous tissues, with 36 healthy and 53 tumour tissues in total. The combined outcomes unveiled 235 generally changed genes, including 83 upregulated and 152 downregulated DEGs, that were obviously expressed in LSCC tumour specimens (p < 0.05, |logFC| > 2). According to the DEGs to bioinformatics analysis, containing GO enrichment, KEGG pathway, PPI network and survival analyses, unveiled that genes related to LSCC and pathways may play a vital function in the cancer initiation and development.

In the GO term enrichment analysis, the DEGs were importantly connected with the terms ‘ECM organization,’ ‘extracellular region’ and ‘heparin binding.’ KEGG pathway analysis indicated that the roles of the DEGs were enriched in ‘ECM-receptor interaction,’ ‘drug metabolism-cytochrome P450,’ ‘focal adhesion,’ ‘PI3K-Akt signalling pathway’ and ‘complement and coagulation cascades.’ The ECM is a sophisticated and dynamic molecular network that surrounds tumour cells and plays vial functions in tumour progression and metastasis (Kim et al., 2016; Xu, Zhang & Zhao, 2017). As tumour cells proliferate, the surrounding ECM experiences important architectural alterations through a dynamic interplay between the microenvironment and resident cells (Grossman et al., 2016). In addition, a recent study recorded that cytochrome P450 inhibited the activity of the metabolic enzymes CYP2C9*2 and CYP2C9*3, which could directly control tumorigenesis by reducing epoxyeicosatrienoic acid production (Hunter et al., 2015; Katiyar et al., 2017). The PI3K-Akt signalling pathway has a vital function in LSCC by suppressing cell death (Chrysovergis et al., 2019; Yang et al., 2019). P53, a tumour suppressor factor, initiates DNA repair, cell cycle arrest and apoptosis and reacts to various kinds of cancer therapies (Cui, Qu & Liu, 2019; Ragos et al., 2018). The changes above are connected with the findings in this study that the BPs of the DEGs necessarily contribute to the progression of LSCC.

Furthermore, analysis showed that the most significant module from the PPI network of LSCC DEGs was connected with the cell cycle and cell metabolism. After further analysis of the DEG PPI network, nine hub genes, MMP9, SPP1, SERPINE1, MMP1, MMP13, MMP3, CXCL8, OSM and COL1A1, were identified, and were all obviously upregulated in LSCC tissue compared with healthy tissues. In addition, the UALCAN and GEPIA online resources were requested for further confirmation of the expression levels of these key genes in LSCC. Each database clearly demonstrated the same tendency in expression as indicated by bioinformatics analysis. Moreover, applying the data from GEPIA, it was recorded that people with LSCC with highly expressed SPP1, SERPINE1 and MMP1 had worse survival outcomes.

Matrix metalloproteinases, which are important proteolytic enzymes, can degrade almost all ECM components and are closely related to tumour infiltration, invasion and metastasis (Grzelczyk, Szemraj & Józefowicz-Korczyńska, 2016). Our research shows that MMP9, MMP1, MMP13 and MMP3 are highly expressed in LSCC. In addition, a few studies have focused on the correlation between LSCC and MMP expression levels. The upregulation of MMP9 and MMP3 has been connected to metastatic progression (Matulka et al., 2019; Zhou & Qi, 2015). The increase in MMP3 levels may has a relation with the regulation of the ERK/MAPK signalling pathway by placental growth factor (Zhou & Qi, 2014). Recent works have indicated that MMP9 increases tumour resistance to anti-PD-1 antibodies (Zhao et al., 2018). A study by Krecicki et al. (2003) showed that MMP1 and MMP13 were highly expressed in LSCC patients. Other studies demonstrated that MMP1 is highly expressed in LSCC and correlates with patient prognosis, which is consistent with our conclusions (Kalfert et al., 2014; Liu et al., 2011). In our work, MMP1 was proven to be highly expressed in LSCC cells, and inhibition of MMP1 suppressed LSCC cell proliferation, invasion and migration. These data indicate that the inhibition of MMP1 might be a bright measure to treat LSCC. The increased expression of MMP13 is also associated with metastatic progression. Immunohistochemical experiments have shown the correlation of MMP13 with TIMP1 (tissue inhibitor of MMPs), which is important in the progression of LSCC (Culhaci et al., 2004). MMPs play vital roles in the spread of malignant tumours by modulating local tumour cell invasion, distant metastasis, angiogenesis and apoptosis (Rydlova et al., 2008). Accordingly, MMP inhibitors are still hopeful for the cure of LSCC.

SPP1, also known as osteopontin, is one of the most significantly overexpressed genes in LSCC and is closely related to LSCC progression. SPP1 is a multifunctional gene that was first reported as a biomarker in the cell epithelial transformation process (Han et al., 2019). However, reports of SPP1 in laryngeal diseases, especially LSCC, are insufficient. In cancer research, the published literature has provided a basic outline of SPP1 biofunctions in tumorigenesis and processes. In colorectal cancer, high SPP1 expression is correlated with poor survival, high TNM stage and positive venous invasion (Assidi et al., 2019). SPP1 participates in the recurrence and metastasis of prostate cancer by mediating the BP of the Smad4/PTEN pathway (Ding et al., 2011). Additionally, SPP1 expression regulation can promote cell growth and mobility in ovarian cancer, and this course may has the relation with the β1/FAK/AKT pathway (Zeng et al., 2018). As the main regulator of the plasminogen activator system, SERPINE1 functions prominently in controlling tumour cell migration and ECM remodelling (Pavón et al., 2016). Furthermore, SERPINE1 could induce the epithelial-to-mesenchymal transition (EMT) process and improve tumour cell survival in ovarian and breast cancers (Azimi et al., 2017; Pan et al., 2017). In this work, bioinformatics analysis revealed obviously increased expression levels of SPP1 and SERPINE1 in LSCC tissues, which led to poor clinical outcomes. We also proved that SPP1 and SERPINE1 were highly expressed in LSCC cells and that LSCC cell proliferation, invasion and migration were suppressed when SPP1 and SERPINE1 were knocked out.

C-X-C motif chemokine 8 (C-XCL-8), recognized as interleukin-8, is a proinflammatory cytokine that plays as a chemotactic factor, mainly for leukocytes (Waugh & Wilson, 2008). Although the specific mechanisms underlying CXCL8-mediated cancer progression may be diverse, CXCL8 has been identified to participate in various cancers (Shrivastava et al., 2014). Our results also verify that highly expressed CXCL-8 can enhance the tumorigenesis and invasion of LSCC. Oncostatin M (OSM), a member of the inflammatory gp130 cytokine family, has been contained to be involved in cancer invasion and metastasis (West et al., 2017). Although limited studies have addressed the role of OSM in LSCC, OSM, as a pleiotropic cytokine, has been shown to function in a variety of cancer cells in vitro, specifically to (1) promote a stem cell-like phenotype and EMT (Junk et al., 2017; West, Murray & Watson, 2014); (2) induce the expression of hypoxia inducible factor-1α, VEGF, and other proangiogenic factors (Fossey et al., 2011; Vollmer et al., 2009); and (3) promote tumour cell invasion and metastasis (Holzer et al., 2004). collagen type I alpha 1 (COL1A1), a major element of the ECM and connective tissues, has been discovered to be actively associated with tumour size and depth of invasion in gastric cancer (Yu et al., 2020). Additionally, research on oesophageal squamous cell carcinoma indicated that COL1A1 might be of vital importance in migration, invasion and progression, and its function may be mediated via the PI3K/Akt/mTOR pathway, p53 pathway, apoptotic pathway and cell cycle (Li et al., 2019).

Conclusions

In summary, bioinformatics analysis identified hub genes and pathways that may play central roles in the occurrence, development and prognosis of LSCC. MMP9, SPP1, SERPINE1, MMP1, MMP13, MMP3, CXCL8, OSM and COL1A1, the hub genes of LSCC, may serve vital functions in the diagnosis and treatment of LSCC, and SPP1, SERPINE1 and MMP1 may be linked with poor prognosis in LSCC patients. We demonstrated that SPP1, SERPINE1 and MMP1 were upregulated in LSCC cells and related to LSCC cell proliferation, invasion and migration.

Supplemental Information

Supplemental Information 1 GO analysis of all differential genes.

Click here for additional data file.

Supplemental Information 2 Data set and analysis of data sets.

Click here for additional data file.

Supplemental Information 3 KEGG analysis of all differential genes.

Click here for additional data file.

Supplemental Information 4 KEGG analysis of all modules.

Click here for additional data file.

Supplemental Information 5 Raw data.

Click here for additional data file.

Supplemental Information 6 Comments for the raw data.

Click here for additional data file.

Supplemental Information 7 GO analysis of all modules.

Click here for additional data file.

Supplemental Information 8 The date of CCK8.

Click here for additional data file.

Supplemental Information 9 The original picture of Western Blot experiment results.

Click here for additional data file.

Supplemental Information 10 The date of transwell.

Click here for additional data file.

Supplemental Information 11 The date of PCR.

Click here for additional data file.

Supplemental Information 12 T he date of cell wound scratch assay.

Click here for additional data file.

Additional Information and Declarations

Competing Interests

Author Contributions

Microarray Data Deposition

Data Availability

The authors declare that they have no competing interests.

Jinhua Ma conceived and designed the experiments, performed the experiments, analyzed the data, prepared figures and/or tables, authored or reviewed drafts of the paper, and approved the final draft.

Xiaodong Hu conceived and designed the experiments, performed the experiments, analyzed the data, prepared figures and/or tables, authored or reviewed drafts of the paper, and approved the final draft.

Baoqiang Dai conceived and designed the experiments, performed the experiments, analyzed the data, prepared figures and/or tables, authored or reviewed drafts of the paper, and approved the final draft.

Qiang Wang conceived and designed the experiments, performed the experiments, analyzed the data, prepared figures and/or tables, authored or reviewed drafts of the paper, and approved the final draft.

Hongqin Wang conceived and designed the experiments, performed the experiments, analyzed the data, prepared figures and/or tables, authored or reviewed drafts of the paper, and approved the final draft.

The following information was supplied regarding the deposition of microarray data:

Data are available at NCBI GEO: GSE59102 and GSE107591.

The following information was supplied regarding data availability:

The data are available in the Supplemental Files.

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
