# Peer review of "Bioinformatics analysis of laryngeal squamous cell carcinoma: seeking key candidate genes and pathways"

_PeerJ, doi:10.7717/peerj.11259_

## Round 0.1 · original submission · Minor Revisions

Dear Dr. Wang,

The manuscript has been reviewed favorably by all the reviewers. Please make the additional changes suggested by reviewers 1 and 2 and we should be able to accept the manuscript for publication if you make these additional changes comprehensively.

Sincerely,

Sonal Choudhary

Reviewer 1 ·

Basic reporting

1) Clear English used.
2) Raw data reference from GEO has been cited.
3) The authors have done a nice work on giving a background introduction of their research
4) Each of the results section is supplemented with articles as per standard

Experimental design

The research question is defined, but bioinformatics methods applied is not presented in details. Please see my comments below
1) While analyzing 2 available microarray dataset, how was the batch effect dealt with? Along with GEO2R methods used, could you please use other tools (such as combat) to apply the batch correction and see how differential expression is changed as compared to results presented

2) For the PPI network modules, how many genes were used for GO analysis.

3) Please explain the PPI methods in details in the method section

4) Legend in figure 3 is too small to read. Please increase the legend size

Validity of the findings

The experimental results support the hypothesis which authors are presenting. But bioinformatics methods used are not robust and needs more detailed analysis.

·

Basic reporting

1. English need to be improve throughout the manuscript.
2. Most relevant references need to add.

Experimental design

Primary research are with in the Aims and scope of the journal.

Research question well defined.

Investigation need to be required more technologically sound.

Methods are fine.

Validity of the findings

Impact is ok.

All the data included.

Conclusions are rightly stated.

Additional comments

Following concerns need to address before this manuscript gets published.
1. Over all English need to improve and most relevant references need to add.
2. All the figures size should be same.
3. Figure 2A text is not readable. Need to improve the figure.
4. Figure 3A text size is too small.
5. Figure 4 there is no statistical significance symbol but author mentioned P value<0.05 in the legend. Moreover its not clear that significance shown between which groups.
6. In Figure 5B, ratio calculation with beta actin is incorrect as per Y axis label. Moreover Panel A has 2 bands so author need to indicate which band is relevant. If considering thick band (upper) is the relevant band then its totally incorrect ratio calculation.

Reviewer 3 ·

Basic reporting

The language can be made better

Experimental design

Solid. No issues.

Validity of the findings

Conclusions are well supported by the analysis. They authors have been very cautious.

Additional comments

This is a manuscript by Ma et al that looks at differentially expressed genes (DEGs) in Laryngeal squamous cell carcinoma (LSCC). The authors find 235 DEGs, including 83 upregulated, 152 downregulated and 9 hub genes. This manuscript is a step forward in the framing of policies for LSCC diagnosis. It is written in a very simple and easy to follow manner. The conclusions are supported by the analysis. This is one of the rare manuscripts that can be accepted as is at PeerJ- a few minor comments

1) The figure legends should be expanded
2) Please check for grammar

---

## Round 0.2 · accepted · Accept

Dear Dr. Wang,

I reviewed your manuscript and I deem it acceptable in PeerJ.